# Metal Ion Periplasmic-Binding Protein YfeA of *Glaesserella parasuis* Induces the Secretion of Pro-Inflammatory Cytokines of Macrophages via MAPK and NF-κB Signaling through TLR2 and TLR4

**DOI:** 10.3390/ijms23179627

**Published:** 2022-08-25

**Authors:** Zhen Yang, Xinwei Tang, Kang Wang, Ke Dai, Yung-Fu Chang, Senyan Du, Qin Zhao, Xiaobo Huang, Rui Wu, Qigui Yan, Sanjie Cao, Yiping Wen

**Affiliations:** 1Research Center of Swine Disease, College of Veterinary Medicine, Sichuan Agricultural University, Chengdu 611130, China; 2Department of Population Medicine and Diagnostic Sciences, College of Veterinary Medicine, Cornell University, New York, NY 14850, USA

**Keywords:** *Glaesserella* *parasuis*, YfeA, C57BL6 mouse, cytokines, toll-like receptors, signal pathways

## Abstract

The *YfeA* gene, belonging to the well-conserved ABC (ATP-binding cassette) transport system *Yfe*, encodes the substrate-binding subunit of the iron, zinc, and manganese transport system in bacteria. As a potential vaccine candidate in *Glaesserella parasuis*, the functional mechanisms of YfeA in the infection process remain obscure. In this study, vaccination with YfeA effectively protected the C56BL6 mouse against the *G. parasuis* SC1401 challenge. Bioinformatics analysis suggests that YfeA is highly conserved in *G. parasuis*, and its metal-binding sites have been strictly conserved throughout evolution. Stimulation of RAW 264.7 macrophages with YfeA verified that toll-like receptors (TLR) 2 and 4 participated in the positive transcription and expression of pro-inflammatory cytokines IL-1β, IL-6, and TNF-α. The activation of TLR2 and TLR4 utilized the MyD88/MAL and TRIF/TRAM pairs to initiate TLRs signaling. Furthermore, YfeA was shown to stimulate nuclear translocation of NF-κB and activated diverse mitogen-activated protein (MAP) kinase signaling cascades, which are specific to the secretion of particular cytokine(s) in murine macrophages. Separate blocking TLR2, TLR4, MAPK, and RelA (p65) pathways significantly decreased YfeA-induced pro-inflammatory cytokine production. In addition, YfeA-stimulated RAW 264.7 produces the pro-inflammatory hallmark, reactive oxygen species (ROS). In conclusion, our findings indicate that YfeA is a novel pro-inflammatory mediator in *G. parasuis* and induces TLR2 and TLR4-dependent pro-inflammatory activity in RAW 264.7 macrophages through P38, JNK-MAPK, and NF-κB signaling pathways.

## 1. Introduction

*Glaesserella parasuis*, belonging to the family Pasteurellaceae, is the causative agent of Glässer’s disease (GD) in swine [1]. The clinical signs of GD are pneumonia, meningitis, polyarthritis, and fibrinous polyserositis, leading to significant economic losses in the pork industry throughout the world [2,3]. To date, over 15 serotypes have been described in this species (20% of isolates have no serotype classification) [4]. Due to the lack of cross-protection among different serotypes, the current regimen of preventing GD by immunization with inactivated vaccines is weak, and treating the disease leans heavily on antibiotic therapy. Accordingly, the development of potent vaccines and therapeutic drugs would be of critical importance.

Bacterial ATP-binding cassette (ABC) transporters play central roles in drug resistance, nutrient uptake, virulence, and possible immunogens [5]. They are suitable targets for the development of antibacterial vaccines and therapies for numerous pathogens [6]. Efficient iron acquisition systems are critical for the ability of bacteria to infect, spread, and grow in mammalian hosts. Iron sequestering is crucial for innate host immune defense against invading pathogens [7]. Many bacteria have a wide variety of transport systems for the acquisition of iron and/or heme, such as Yfe and Feo transport systems [7]. The Yfe system, encoding by the *yfe* gene cluster, is an ABC (ATP-binding cassette) transporter for the transmembrane delivery of iron, zinc, and manganese with a typical periplasmic binding protein (YfeA), an ATPase (YfeB), two inner membrane (IM) permeases (YfeC and YfeD), and a possibly unnecessary protein (YfeE) [8,9]. Biochemical analysis indicates that the Yfe system complements their respective transporters Feo to transport iron. Due to its vital role in metal ion-binding, YfeA has been an essential virulence factor of many bacterial species. During the infection process, *Salmonella enterica* serovar Typhimurium induces a series of virulence genes, including YfeA, that circumvent host defenses and/or acquire nutrients [10]. It has been approved that the Yfe-deletion mutant in *S.* Typhimurium is significantly attenuated in BALB/c mice [11]. In *Yersinia pestis*, the etiological agent of plague, the Yfe system may function effectively to accumulate iron during different stages of the infection process of bubonic plague, demonstrating that YfeA is vital for full virulence of *Y. pestis* during its infection of the host [8]. The main virulence factors may also work well as potential immunogenic antigens such as lipopolysaccharides (LPS) of *E. coli*, and APX toxin of *A. pleuropneumoniae* [12,13,14]. Importantly, YfeA was identified as a potent immunogenic protein in *G. parasuis* by the reverse vaccinology strategy, and it showed cross-reactions when tested with serum raised against serovars 4 and 5 of *G. parasuis* [15]. Nevertheless, the exact biological function of YfeA in *G. parasuis* infection progress has not been well delineated, and the signal pathways mediated by YfeA are yet unexplored.

Innate immunity is the first line of the host’s defense against invading pathogens. Unlike adaptive immunity, innate immunity can quickly activate the immune response system to cope with the foreign immunogen, occupying a unique role in bacterial invasion. Macrophages are the key innate immune cells and are primarily distributed in all major organs [16], and have been considered the critical target cells of numerous immune regulatory systems [17]. Macrophages possess a fine-tuned immune recognition system against extracellular stimulus signaling molecules as sentinel cells. Ample evidence exists to show that microbial sensors of macrophages utilize pattern recognition receptors (PRRs) to detect pathogen-associated molecular patterns (PAMPs), which are commonly conserved in microbial products but are not present in the host itself [18]. At this stage, macrophages are activated, which is characterized by the enhancement of phagocytosis, the acceleration of cytotoxic molecule production, such as reactive oxygen (ROS) and nitrogen species (NO), and the secretion of various cytokines (IL-1β, IL-6, TNF-α) [19,20]. The pro-inflammatory cytokines secreted from macrophages recruit several types of immune cells, including T cells and leukocytes, to the site of infection to eliminate invading microorganisms. Therefore, macrophages have potent microbicidal activity and take part in the host’s defense against invading pathogens [21]. All the above characteristics make TLR ligands potential vaccine candidates or therapeutic targets for a specific pathogen.

In our study, we concentrate on the immunoprotection of YfeA in *G. parasuis*. First, the immunoprotective effects of recombinant YfeA (rYfeA) were evaluated in C57BL6 mice. Then RAW 264.7 were treated with *G. parasuis* YfeA protein, and RT-PCR and ELISA were performed to analyze the expression levels of various pro-inflammatory cytokines. We also analyzed the MAPK and NF-κB signal pathways induced by YfeA on RAW 264.7 and the TLRs and adaptors involved in this process. Moreover, YfeA can trigger the production of pro-inflammatory markers and reactive oxygen species. This study aims to confirm the mechanism of the induction of innate immunity within murine models and further determine the mechanisms of an inflammatory response to YfeA protein exposure at the cellular level, which hopefully lays a foundation for a follow-up development of a subunit or chimeric vaccine in *G. parasuis*.

## 2. Results

### 2.1. Bioinformatics Analysis of YfeA

Phylogenetic analysis showed the *yfeA* gene cluster was widespread in several representative species of bacteria, including *G. parasuis*, *Yersinia pestis*, *Actinobacillus pleuropneumoniae*, *Haemophilus influenzae*, and *Pasteurella multocida* (Figure 1A). Blast analysis showed that YfeA of *G. parasuis* belong to one sole clade, suggesting that this protein is highly conserved in the species, which endows this protein with the qualification to serve as a possible candidate for a subunit vaccine against this species.

The crystal structure of holo YfeA of *Y.pestis* has been determined, which shows the prototypical SBP c-clamp fold with a primary metal-binding site located in the c-clamp arch (Figure 1A,B). YfeA shares 63.7% identity in deduced amino acid sequence between *G. parasuis* and *Y. pestis*. The model of YfeA was predicted by homology-modeling server SWISS-MODEL; then it was aligned with the structure of YfeA of *Y.pestis*. As outlined in Figure 1C, their protein structures are identical. Amino acid conservation plots of YfeA were obtained using the ConSurf server. The result indicated that the metal-binding site was highly conserved across evolution among the bacteria (Figure 1B), including two histidines, one aspartic acid, and one glutamic acid. As shown in Figure 1C,D, the binding site of YfeA of *G. parasuis* interacts with iron and zinc by metal complexation.

### 2.2. Recombinant YfeA Protein Could Be Applied to Cell Treatment

YfeA protein was used as an antigen to investigate its pro-inflammatory effect on macrophages. YfeA protein was expressed as a fusion protein in an active form with 6 × His tags on its N-terminal and purified using Ni-affinity chromatography in vitro. To preclude the interference effect of imidazole’s cytotoxicity, the products were further dialyzed with PBS to remove contamination of imidazole. Then the products were detected using SDS-PAGE and Western blotting. One highly purified band with a molecular mass of approximately 36 kDa, corresponding to the predicted molecular mass of YfeA protein plus 6 × His tags, was obtained (Figure 2A). Western blotting (WB) results verified the band was YfeA by using its specific antibody (Figure 2B), demonstrating that the YfeA protein was successfully expressed in BL21 (DE3). After endotoxin (LPS) removal, the concentration of endotoxins (0.1 EU) was less than 0.3 EU (endotoxin unit) per milliliter in YfeA, which could be applied to treating macrophages (data not shown).

### 2.3. The Immunity of rYfeA-Vaccinated Mice to G. parasuis

We initially assayed the immunoprotective effect of rYfeA in the mice model, before validating its immune protective effect in piglets. The rYfeA+adjuvant group displayed a mortality rate of 40% (4/10) upon challenge with highly virulent *G. parasuis* SC1401, while the rYfeA-alone/PBS/adjuvant-alone groups showed mortality rates of 100% within 48 h (Figure 3A). In the passive immunization assay, all the mice receiving antisera elicited by rYfeA alone/PBS/adjuvant alone died after the challenge with 1 × 10 ^8^ CFU SC1401 within 36 h. Forty percent of the mice (4/10) immunized passively with antisera from rYfeA mixed with adjuvant were prevented from the lethal *G. parasuis* challenge (Figure 3B), and nearly all survivors recovered within 72 h. In the active immunization assay, the murine lungs were collected for further histopathologic examaination to detect severity levels of the pathological lesion after the mice were humanely euthanized. As illustrated in Figure 3C, pathological lesions occurred in all immunized groups, but mice in the rYfeA+adjuvant group displayed less severe pathological damage than the other four treated groups. In the rYfeA+adjuvant group, lung sections showed slightly thickened alveolar walls and mild pulmonary lymphocytic infiltration. In addition, a few granulocytes in the major lumen of the blood vessels, mild hemorrhage of the local bronchial, and small amounts of red blood cells in the lumen of the bronchial were also observed. By contrast, in the adjuvant-alone group, lung sections exhibited severe alveolar wall thickening and deep staining or fragmentation accompanied by a mass of lymphocytic infiltration. Meanwhile, large amounts of red blood cells in the blood vessels’ lumen, and hemorrhaging were also observed. No anomalies were observed in the mock group. In general, YfeA has the potential to become a subunit candidate through mouse challenge experiments.

### 2.4. YfeA Actively Induces the Secretion of Pro-Inflammatory Cytokines IL-1β, IL-6, and TNF-α

Macrophages can be stimulated to secrete pro-inflammatory cytokines by their interaction with microbes and microbial products. To analyze the function of the YfeA antigen in bacterial pathogenesis, pro-inflammatory cytokine responses were measured on RAW 264.7 macrophages. The RAW 264.7 cell line is monocytes and macrophages derived from ascites of the Abelson murine leukemia virus-induced tumor model and are considered one of the best models for macrophages concerning their ability to perform pinocytosis, phagocytosis, and antigen presentation [22]. During the study, we preliminarily tested IL-1β, IL-6, and TNF-α using ELISA and qRT-PCR to verify the variable levels of pro-inflammatory cytokines secreted by RAW 264.7 cells. After treatment with YfeA of various concentrations (10, 20, 40, and 60 μg/mL), the secretion levels of all three cytokines were significantly upregulated in contrast with the negative control (*p* < 0.05), in which only low levels of TNF-α, IL-1β, and IL-6 were detected. While in the positive control, either 100 ng/mL or 200 ng/mL LPS could dramatically elevate the secretion levels of IL-1β, IL-6, and TNF-α. A total of 100 ng/mL LPS was sufficient for macrophages to produce high levels of cytokines, so the difference between 100 and 200 ng/mL is not obvious. Consistent with the secretion levels, the transcriptional levels of IL-1β, IL-6, and TNF-α also augmented substantially following YfeA and LPS stimulation, in contrast with the negative control (*p* < 0.05) (Figure 4), indicating the activation of pro-inflammatory activity of RAW 264.7. In addition, the secretion and transcriptional levels of IL-4, an anti-inflammatory cytokine, were tested. As illustrated in Figure 4D, the transcriptional level of IL-4 merely fluctuated within a statistically insignificant range, despite the cells being treated with LPS or a low dose of YfeA (10 μg/mL). In contrast, a high-dose YfeA (20, 40, or 60 μg/mL) could slightly elevate its transcriptional level. However, the improvement extent of IL-4 was much lower than that of IL-1β, TNF-α, and IL-6, and there was no significant difference in the secretion levels of IL-4 among the three groups (*p >* 0.05), indicating that YfeA actively induces the secretion of pro-inflammatory rather than anti-inflammatory cytokines. Furthermore, 20, 40, and 60 µg/mL of YfeA treatment triggered a statistically significant increase in the transcriptional level of IL-6 and TNF-α compared to 10 µg/mL of YfeA, demonstrating a positive correlation between YfeA concentration and cytokine induction levels (data not shown). These observations signify that YfeA actively induces the secretion of pro-inflammatory cytokines IL-1β, IL-6, and TNF-α.

### 2.5. YfeA Stimulates Reactive Oxygen Species (ROS) Production in RAW 264.7 Cells

ROS are essential signaling molecules for initiating the host inflammatory response and play an important role in cell signal transduction [23]. The FCM assay was conducted to evaluate reactive oxygen species (ROS) levels, which is one of the critical pro-inflammatory hallmarks of macrophages [24]. In the FCM assay, the fluorescence probe DCFH-DA (final concentration of 5 μM) was used to treat the RAW 264.7 macrophages that received YfeA, LPS (positive control), or PBS (negative control) in advance. Only low background noise was presented in the mock group, and 12.4% of cell population input was sorted as ROS-positive, showing a basic level of ROS in these cells. In comparison, this percentage significantly augmented after cells had been treated with endotoxin-deprived YfeA (25.9% or 40.3%), a similar growth trend to the cells treated with LPS (31.7% or 60.6%) (Figure 5). These results established that YfeA could actively induce ROS production in macrophages.

### 2.6. TLR2 and TLR4 Play a Regulatory Role in the Secretion of Pro-Inflammatory Cytokines in Macrophages

WB and blocking assays were adopted to identify the cell surface PRRs of YfeA. We explored if YfeA protein could induce overexpression of TLR2 and/or TLR4 using WB analysis to prove our hypothesis. As demonstrated in Figure 6, significantly increased expression levels of TLR2 and TLR4 in the YfeA-treated group were observed compared with the negative control. It is well known that LPS is a ligand of TLR4 rather than TLR2, so the overexpression of TLR2 in the YfeA-treatment group reflects that YfeA acts as an inducer, not an inducer of the residual endotoxin that may be presented.

Next, we blocked TLR2 and TLR4 by using their corresponding anti-TLR2 and anti-TLR4 monoclonal antibodies to establish the receptor specificity of YfeA further. After TLR2 and TLR4 were separately blocked, levels of IL-1β, IL-6, and TNF-α were measured. ELISA results showed that production of all three pro-inflammatory cytokines decreased significantly after TLR4 was blocked compared with the YfeA-sole-treated group, especially the level of TNF-α (*p* < 0.05), which fell by almost half. In comparison, only the secretion level of IL-6 decreased significantly when TLR2 was blocked in comparison with the YfeA treatment, while the other two cytokines declined slightly (*p* > 0.05). In summary, TLR4 and TLR2 mediated YfeA-induced secretion of pro-inflammatory cytokines in macrophages in *G. parasuis* infection, and TLR2 seems to be more likely to mediate the secretion of IL-6 in the process.

### 2.7. YfeA Induces Upregulation and Concentration of the Adaptor Protein MyD88 and TRIF in Macrophages

TLR2 utilizes the adaptor protein MyD88/MAL pair to initiate TLR signaling, while TLR4 utilizes the MyD88/MAL and TRIF/TRAM pair [25]. To further corroborate the initiation of the TLR signal, we performed an IF assay to determine whether YfeA could trigger a change in the levels of the adaptor protein MyD88 and TRIF in macrophages. As outlined in Figure 7, the fluorescence of MyD88 (red light) and TRIF (green light) was dim and diffuse in the control group. After RAW 264.7 cells were treated with YfeA or LPS, the fluorescence of MyD88 and TRIF became bright and concentrated, and both LPS and YfeA increased the expression of MyD88 and TRIF (*p* > 0.05). The results showed the activation of TLR2 and TLR4 initiated TLRS signaling by upregulating and accumulating the levels of adaptor molecule MyD88 and TRIF in macrophages.

### 2.8. YfeA Activates Both MAP Kinases and NF-κB Signal Pathways of RAW 264.7 Cells

To investigate the signal pathways for YfeA engaged in boosting pro-inflammatory cytokines, Western blotting was performed to determine the levels of phosphorylation of MAP kinases, including JNK, ERK1/2, p38 signal molecules, and the RelA (p65) signal molecule belonging to the NF-κB pathway. As outlined in Figure 8, RAW 264.7 cells induced by YfeA stimulation augmented the phosphorylation level of JNK, ERK1/2, and p38 significantly when compared to untreated cells (*p* < 0.05), and the expression level of p-p65 increased slightly (*p* > 0.05). Hence, the results indicated that YfeA activates the p38, JNK1/2, and ERK1/2 MAP kinase.

### 2.9. YfeA Induces Pro-Inflammatory Cytokines Production through MAPKs and NF-κB Signal Pathways

To characterize the details of signal pathways stimulated by YfeA, we used the inhibitors SP600125 (JNK-MAPK inhibitor), U0126 (Erk1/2-MAPK inhibitor), SB203580 (p38-MAPK inhibitor), and PDTC (NF-κB inhibitor) to block their corresponding signal pathway in RAW 264.7, which was followed by collecting their supernatants to quantify the production of IL-1β, IL-6, and TNF-α. The inhibitors used in this study refer to previous studies [26]. As shown above, we treated RAW 264.7 cells with 10 µM of each inhibitor before detecting the levels of IL-1β, IL-6, and TNF-α in the cell supernatants. As shown in Figure 9, when the JNK-MAPK and NF-κB pathways were blocked, the levels of all three cytokines tested decreased when compared with the YfeA-induced group, especially the level of IL-6 in the JNK-MAPK pathway-blocking group, which declined from 484.58 ± 0.5 pg/mL to 24.16 ± 4.4 pg/mL. The three cytokines triggered by YfeA protein seem to be produced through different signaling pathways. IL-1β and TNF-α mainly depend on the JNK-MAPK and NF-κB signaling pathways, while IL-6 appears to be dependent on p38-MAPK, ERK, JNK-MAPK, and NF-κB signaling pathways.

## 3. Discussion

*G. parasuis* is usually a benign swine commensal in the upper respiratory tract but may cause severe multiorgan dysfunction and vascular lesions when the host immunity drops [27]. Commercial bacterins are usually used to vaccinate swine against *G. parasuis*, but the lack of cross-reactivity makes sole-serotype inactivated vaccine an invalid means of protection against this disease [28].

YfeA, the metal-binding protein, was identified as an in vivo-induced antigen during bubonic and septicemic plague infection [29], and was reported to be an immunogenic protein in *G. parasuis* and showed cross-reactions when tested with serum raised against serovars 4 and 5 [30]. Several studies showed that mice, including C57BL6 and BALB/c mice, are an ideal animal model to study the pathogenesis and diagnosis of and immunization against *G. parasuis* infection [31,32]. Additionally, the RAW 264.7 cell lines can phagocytize a variety of invasive pathogenic microorganisms or their components, and while activating lymphocytes or other immune cells, activate the immune response of the body, so it has been widely accepted as an in vitro model to investigate cellular inflammation responses and various TLR signaling studies [33,34]. Thus we chose BALB/c mice and RAW 264.7 cells as infection model of YfeA, prior to validating its immune protectivity in pigs.

In the study, we found that YfeA is highly conserved in *G. parasuis*, and its structure is identical between *Y.pestis* and *G. parasuis*. rYfeA-vaccinated mice obtain some level of immunity to *G. parasuis*. In a word, YfeA could be exploited as a potential subunit candidate for vaccination against *G. parasuis*, prompting us to investigate the specific mechanisms of induction of innate immunity. Using qRT-PCR and ELISA assay, we demonstrated YfeA protein of *G. parasuis* induced the transcription and secretion of pro-inflammatory cytokines, including IL-1β, IL-6, and TNF-α. We utilized the pET-32a-His protein (MW:19 Kda) as a control, and the rYfeA-treated group showed a higher transcription level of pro-inflammatory cytokines (data not shown). Toll-like receptors play a key role in the interactions of hosts and microbes. Western blotting (WB) analysis and toll-like receptor blocking assay showed YfeA is a ligand of TLR2 and TLR4 on RAW 264.7 macrophages. The results were different from a previous finding of PotD protein in *G. parasuis* whose receptor is TLR4 instead of TLR2, which suggested the pro-inflammatory response of the RAW 264.7 is specific for the YfeA protein, and not just due to a general pro-inflammatory response produced by macrophages towards any bacterial protein/LPS in general [26]. As shown in Figure 10, after recognizing YfeA, TLR2 and TLR4 dimerize on the cell membrane. Indirect immunofluorescence assay showed upon receptors dimerization, conformational changes occur in the TIR domain of TLRs, leading to the association of MyD88 with TLR2 and TLR4 and of TRIF with TLR4 via homotypic interaction between their TIR domains. Further study in signal pathways confirmed that both JNK-MAPK and NF-κB play primary roles in the pro-inflammatory response induced by YfeA stimulation in macrophages. In addition, a distinct signaling pathway exists for each cytokine induction, reflecting the functional differences among the cytokines. The present experiments mainly involve the immunity mechanism of RAW 264.7 macrophages; whether YfeA protein is immunogenic to porcine macrophages needs further study. Moreover, further studies concerning adaptive immunity mediated by YfeA and the immunoprotective effect of YfeA on the animal model are needed.

In conclusion, we demonstrated YfeA protein of *G. parasuis* could promote the secretion of pro-inflammatory cytokines. This process involves TLR2 and TLR4 recognition and cell signaling pathways, including P38, JNK-MAPK, and NF-κB. The relationship between YfeA and acquired immunity and the evaluation of its protective effect deserves further study.

## 4. Materials and Methods

### 4.1. Bacterial and Cell Culture Condition

Wild-type *G. parasuis* SC1401 (highly virulent, serotype 5) was grown in tryptic soy broth (TSB, Difco, Franklin Lakes, NJ, USA) supplemented with 5% newborn bovine serum (Solarbio, Beijing, China) and 0.1% (*w*/*v*) nicotinamide adenine dinucleotide (NAD, Sigma-Aldrich, St. Louis, MO, USA). YfeA protein expression strain was cultured in LB broth or LB agar. All strains were shaken at 220 rpm at 37 °C. Mouse monocyte/macrophage leukemia cell line RAW 264.7 were obtained from ATCC and cultured in DMEM (Gibco, Invitrogen, Carlsbad, CA, USA) supplemented with 10% FBS (Gibco, Invitrogen, Carlsbad, CA, USA) at 37 °C and in 5% CO_2_ without any antibiotic. In the follow-up study, we adopted the RAW 264.7 cell line to dissect immune responses and signal pathways. RAW 264.7 cells were plated into 12-well plates at a density of 1 × 10^6^ cells/well and stimulated with YfeA protein at a final concentration of 10, 20, 40, or 60 µg/mL for concentration-dependent tests. To detect immune-responsive cell receptors and signal pathways, the cells were collected at 12 h after stimulation. To exclude the possible lipopolysaccharide (LPS) contamination, polymyxin B (PMB) was added at a final concentration of 10 µg/mL throughout the experiments, except where LPS was utilized as a positive control.

### 4.2. Expression/Purification of Recombinant YfeA Protein and Removal of Endotoxins

Expression of recombinant protein of YfeA was performed using the *E. coli* expression system. Briefly, the *G. parasuis yfeA* gene (GenBank: CP001321.1) minus the 20 amino acid signal peptide sequences were obtained from the genomic DNA of SC1401. YfeA expression primers are listed in Table 1. The PCR products were ligated to the pET-28a (+) vector (HANBIO, Shanghai, China), giving rise to pET-yfeA. Recombinant clone pET-yfeA was induced with 0.5 mM IPTG for 6 h at 37 °C for optimum expression of YfeA, which was purified by Ni-affinity chromatography (Bio-Rad, Hercules, CA, USA). Purified YfeA was dialyzed with 10 L of PBS for 48 days at 4 °C and then confirmed by SDS-PAGE electrophoresis. After the YfeA protein was dialyzed, we used a toxin eraser (Genscript, Piscataway, NJ, USA) to reduce the level of endotoxin contamination.

### 4.3. Immunization and Mouse Challenge

All experimental protocols were authorized by the Animal Ethics Committee of Sichuan Agricultural University and were carried out accordingly. Fifty 6-week-old female C57BL6 mice were randomly assigned to 5 groups of 10 each. A total of 200 mg rYfeA, which was emulsified with water adjuvant MONTANIDE™ GEL 01(total volume: 200 μL), was evaluated in this experiment. Except for the mock group, the four groups were subcutaneously immunized with rYfeA, emulsified rYfeA, adjuvant, and emulsified PBS, respectively. Subsequent booster injections with the same antigen were given on the 14th day. Seven days after the booster immunizations (on the 28th day), serum was obtained from each mouse by tail vein bleeding. Then all mice in the treatment group were intraperitoneally challenged with a lethal dose of wild-type *G. parasuis* strain SC1401 (1 × 10^8^ CFU, 0.2 mL).

After mice had been challenged for 72 h, three animals in each group were humanely euthanized, necropsied and their lungs were collected and treated with 10% neutralized formalin for histological section and H&E staining and examination. To evaluate the protectivity elicited by mouse anti-YfeA antisera, a further passive immunization protective effect was determined. In brief, another fifty 6-week-old mice (10 mice per group) were intraperitoneally injected with 150 µL of pooled serum obtained from the immunized mice 2 h before the challenge with a lethal dose of *G. parasuis* SC1401. The surviving mice were humanely euthanized at 72 hpi. The experiments were independently performed two times.

### 4.4. The Capacity of YfeA to Stimulate Cytokine Production

ELISA and qRT-PCR were performed to investigate the capacity of YfeA to stimulate cytokine synthesis. In brief, RAW 264.7 were incubated with YfeA at a final concentration of 10, 20, 40, or 60 µg/mL. LPS (100 or 200 ng/mL, Solarbio, without polymyxin B) was added to a group of three wells in parallel to serve as a positive control. An equivalent volume of PBS was added to the negative control group. The cultures were incubated for 12 h. Then the supernatants and cells were harvested, respectively.

The levels of IL-1β, IL-4, IL-6, IL-10, and TNF-α were determined in culture supernatants using the double-antibody sandwich ELISA kits (Biotech), following the manufacturer’s instructions. In addition, transcription profiles of IL-1β, IL-4, IL-6, IL-10, and TNF-α were analyzed by qRT-PCR. Primers for qPCR are listed in Table 1. These experiments were independently performed three times in triplicate.

### 4.5. Measurement of ROS Production

After being treated with YfeA (20 and 40 µg/mL) or LPS (100 and 200 ng/mL) for 12 h, RAW 264.7 macrophages were collected and incubated with DCFH-DA (20 μM) in a serum-free medium for 45 min at 37 °C in the dark. The LPS-treatment group and PBS-treatment group were set as positive and negative controls, respectively. These cells were stained with 2′,7′-dichlorofluorescein diacetate (DCFDA), a nonpolar dye that easily diffuses into cells and is deacetylated by cellular esterases to a nonfluorescent derivative DCFH. In the presence of ROS, DCFH can be oxidized to highly fluorescent 2′,7′-dichlorofluorescein (DCF). DCFH-DA staining cells were immediately examined using a flow cytometer (FACS Verse, BD Biosciences). The frequencies of ROS donor cells were analyzed by Flowjo software (Flowjo LLC, Ashland, OR, USA). These experiments were independently performed three times in triplicate.

### 4.6. Toll-like Receptors 2 (TLR2) and 4 (TLR4) Blocking Assay

The macrophage pattern recognition receptors (PRRs) involved in YfeA protein and cell interaction were analyzed using a blocking assay. First, we blocked either TLR2 or TLR4 on RAW 264.7 using their corresponding monoclonal antibodies (mAb) (BioLegend). In comparison, both antibodies were added to a final concentration of 10 µg/mL for 2 h in a parallel group in order to block both receptors to identify the coincidence summing effect. After pretreatment with mAb, YfeA protein was added to the RAW 264.7 cells at a final concentration of 20 µg/mL, then incubated for 12 h. Then cell culture supernatant was collected for routine detection of cytokines IL-1β, IL-6, TNF-α, and IL-4 using ELISA kits. Results were analyzed to determine the cytokine secretion levels and to confirm receptors contributing to the response to YfeA exposure.

### 4.7. Indirect Immunofluorescence (IF) Assay for Detection of Adaptor Protein MyD88 and TRIF

Cells were seeded onto coverslips for treatment with YfeA overnight. After being fixed, the cells were permeabilized and blocked according to the routine procedures. RAW 264.7 cells were then incubated with anti-MyD88 (ABclonal, Wuhan, China) and anti-TRIF (ABclonal) primary antibodies (1:200) overnight at 4 °C. After washing, cells were incubated with FITC-conjugated and CY3-conjugated secondary antibody (1:500) (ABclonal) for 1 h at room temperature. Finally, nuclear staining was done with DAPI.

### 4.8. Western Blotting Assay for Induction of Pro-Inflammatory Cytokines by YfeA

RAW 264.7 macrophages were treated with or without YfeA (20 µg/mL) for 12 h, the total cellular protein was extracted using a Protein Extracting Kit (Solarbio, Beijing, China). Aliquots corresponding to 50 µg of each sample were analyzed using Western blotting (WB). To analyze the YfeA-induced signal transduction residue phosphorylation, the primary monoclonal antibodies, including Erk1/2 (rabbit, 42/44 kDa; 1:2000; Abcam ab184699), p-Erk1/2 (rabbit, p-ERK1: Thr202/Tyr204, p-ERK2: Thr185/Tyr187; 42/44 kDa; 1:2000; Abcam ab278538), p38-MAPK (rabbit, 41 kDa; 1:1000; Abcam ab31828), p-p38-MAPK(rabbit, Tyr182; 41 kDa; 1:500; Abcam ab47363), JNK (rabbit, 48 kDa; 1:1000; Abcam ab179461), p-JNK (rabbit, Tyr185/223; 48 kDa; 1:10,000 Abcam ab76572), p65 (NF-κB; rabbit, 60 kDa; 1:2000; Abcam ab16502), p-p65 (NF-κB; rabbit, Ser536; 60 kDa; 1:5000; Abcam ab86299), TLR2 (rabbit, 89 kDa; 1:500; Abcam ab213676), and TLR4 (rabbit, 89 kDa; 1:500; Abcam ab13556) were applied at this stage.

### 4.9. Effect of Inhibitors on the MAPK and NF-κB Signal Pathways

To evaluate the roles of the ERK1/2, p38, JNK, and NF-κB signal pathways in the YfeA-mediated production of pro-inflammatory cytokines, specific inhibitors against each pathway were used. Prior to inhibiting cell signal pathways by using inhibitor agents SB203580 (p38-MAPK inhibitor), U0126 (Erk1/2 inhibitor), SP600125 (JNK inhibitor), and PDTC (NF-κB inhibitor), the cytotoxicity of these signal pathway inhibitors were evaluated in vitro using a CCK-8 counting assay.

RAW 264.7 cells were pretreated with the above specific inhibitors (10 µM each) in parallel for 45 min prior to YfeA addition. Another DMSO (1 µL/mL, AMRESCO) group was a negative control (NC). After 8 h incubation with YfeA or maintaining the status quo, culture supernatant was collected to determine the production of diverse cytokines using ELISA kits. Results were analyzed to assess cytokine response levels and to confirm the identity of cellular signal pathways contributing to cellular responses to the YfeA exposure.

### 4.10. Statistical Analysis

Phylogenetic trees were made by MEGA v6 and pruned using the online server iTOL v4 (https://itol.embl.de/, accessed on 10 May 2021). The amino acid sequence alignment was done with DNAMAN. Amino acid conservation plots of YfeA protein were obtained using the online software the ConSurf server (https://consurf.tau.ac.il/, accessed on 10 May 2021). The structure of YfeA was predicted using the protein homology-modeling server SWISS-MODEL (https://swissmodel.expasy.org, accessed on 10 May 2021). The interaction between YfeA protein and metal ions was predicted by online software PLIP (protein–ligand interaction profiler). All protein structures were visualized using Pymol. All experiments were performed in triplicate and the data are expressed as mean ± standard deviation. Comparisons of several independent test series were evaluated by analysis of two-way analysis of variance (ANOVA tests). Significant differences between groups are indicated by * *p* < 0.05, ** *p* < 0.01, *** *p* < 0.001, and **** *p* < 0.0001.

## Figures and Tables

**Figure 1 ijms-23-09627-f001:**
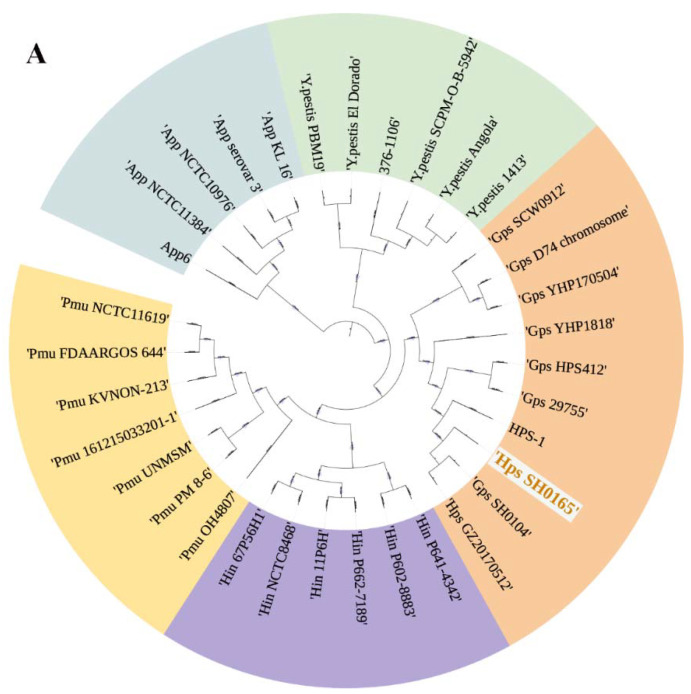
Bioinformatics analysis of YfeA. (**A**) Phylogenetic tree of *YfeA*. Neighbor-joining (NJ) tree demonstrated the overall diversity of YfeA from *G. parasuis* (*Gps* or *Hps*, pink), *Yersinia pestis* (green), *A. pleuropneumonia* (blue), *H. influenzae* (purple), and *P. multocida* (yellow). (**B**) Comparison of YfeA protein structure between *Y. pestis* (white) and *G. parasuis* (gold). (**C**) Amino acid conservation plots of YfeA protein analyzed by ConSurf. Based on the relative degree of conservation, each amino acid site was assigned a value from 1 (variable; light blue) to 9 (conserved; dark purple). A yellow box indicates the metal ion-binding domain of YfeA. Schematic diagram of YfeA of *G. parasuis* binding with zinc (**D**) and iron (**E**). The gray sphere represents zinc; the orange sphere represents iron; the yellow dashed lines represent the metal complexation between the amino acids and metal ions.

**Figure 2 ijms-23-09627-f002:**
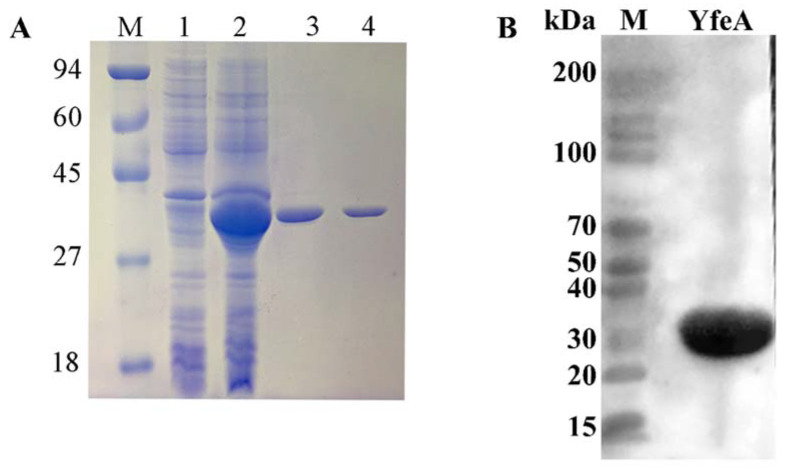
Recombinant YfeA protein could be applied to cell treatment. (**A**) SDS-PAGE analysis of the expression and purification of YfeA. M: protein marker; 1: BL21 containing pET-28a; 2: supernate of BL21 containing YfeA-pET-28a after inducing and disruption; 3: the purified YfeA protein; 4: the purified YfeA protein (dialyzed, ultra-filtrated, and LPS-removed). (**B**) Detection of the specificity of purified YfeA using WB assay. M: protein marker; YfeA: the purified YfeA protein (dialyzed, ultra-filtrated, and LPS-removed).

**Figure 3 ijms-23-09627-f003:**
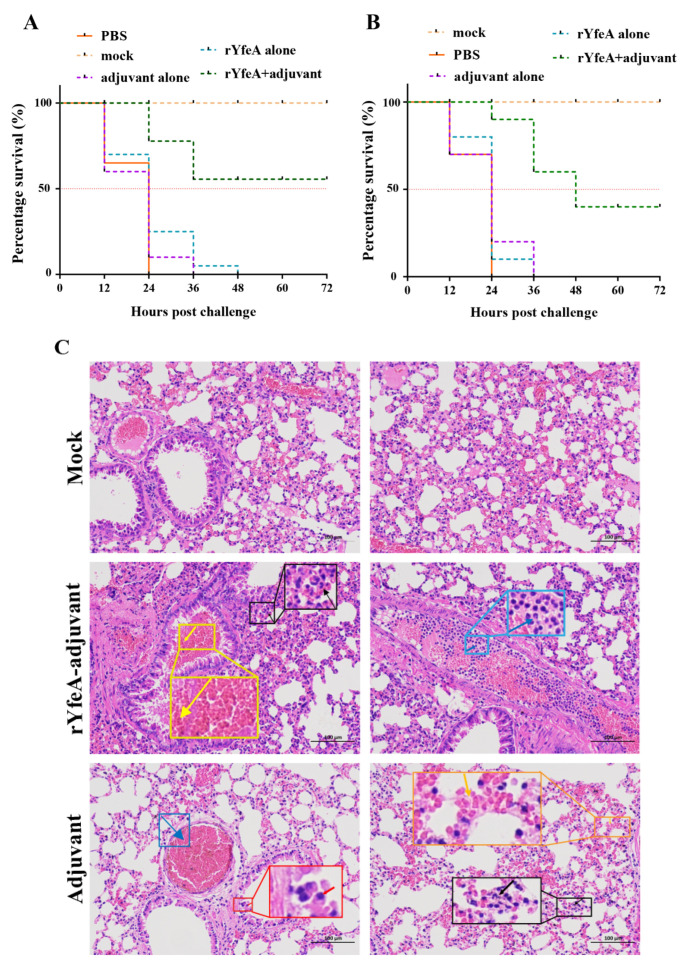
The immunity of rYfeA-vaccinated mice to *G. parasuis*. The survival rate of C57BL6 mice was challenged with a lethal dose (1 × 10^8^ CFU/mouse) of *G. parasuis* SC1401 (**A**,**B**). (**A**) In the active immunization assay, adjuvant-alone/rYfeA-alone/PBS groups engendered 100% (10/10) death in mice model within 48 h post-injection, while the rYfeA+adjuvant group caused 40% (4/10) of death within 36 h. (**B**) In the passive immunization experiment, four mice (4/10) immunized passively with antisera from rYfeA+ adjuvant mixture were protected from the lethal *G. parasuis* challenge. All the mice acquiring antisera stimulated by adjuvant-alone/rYfeA-alone/PBS died after the challenge with *G. parasuis* SC1401 within 36 h. The mock group presents mice that were neither vaccinated nor challenged with SC1401, while the PBS group indicates mice receiving both PBS inoculation and bacteria injection. All assays were carried out at least two times. Histopathologic analysis of mice lungs (**C**). Lung tissues were collected from mice in different groups at 72 hpi, and used for hematoxylin and eosin (H&E) staining and histopathological examination. Mock group (Mock): no obvious pathological lesion. rYfeA-adjuvant group: mild thickening of alveolar walls is widely seen, scattered granulocytes infiltration (black arrow); local hemorrhage and few tiny red blood cell extravasations (yellow arrow); few granulocytes infiltration (blue arrow). Adjuvant group: a large number of bronchial epithelial cells were exfoliated (black arrow); severe congestion in the blood vessels (blue arrow); more alveolar walls were found to be slightly thickened and more scattered granulocyte infiltration (black arrow); the congestion of numerous alveolar wall capillaries (orange arrow).

**Figure 4 ijms-23-09627-f004:**
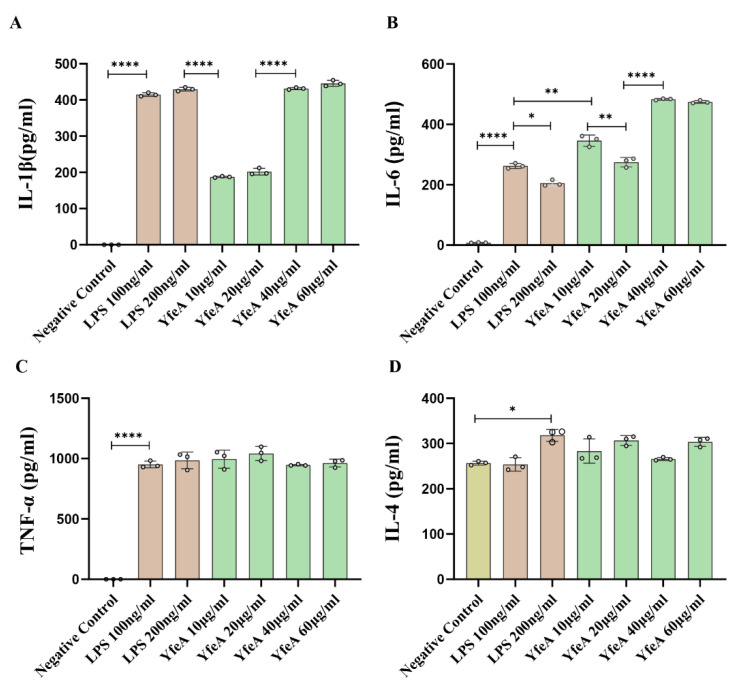
Detection of the levels of pro-inflammatory cytokines induced by YfeA. RAW 264.7 macrophages were incubated with 10, 20, 40, and 60 µg/mL of YfeA protein and 100 or 200 ng/mL LPS (positive control) overnight, as well as single culture media (negative control). ELISA detection of the secretion levels of IL-1β (**A**), IL-6 (**B**), TNF-α (**C**), and IL-4 (**D**). All assays were performed three times independently. Error bars represent standard deviations. Significant differences between groups are indicated by * *p* < 0.05, ** *p* < 0.01 and **** *p* < 0.0001.

**Figure 5 ijms-23-09627-f005:**
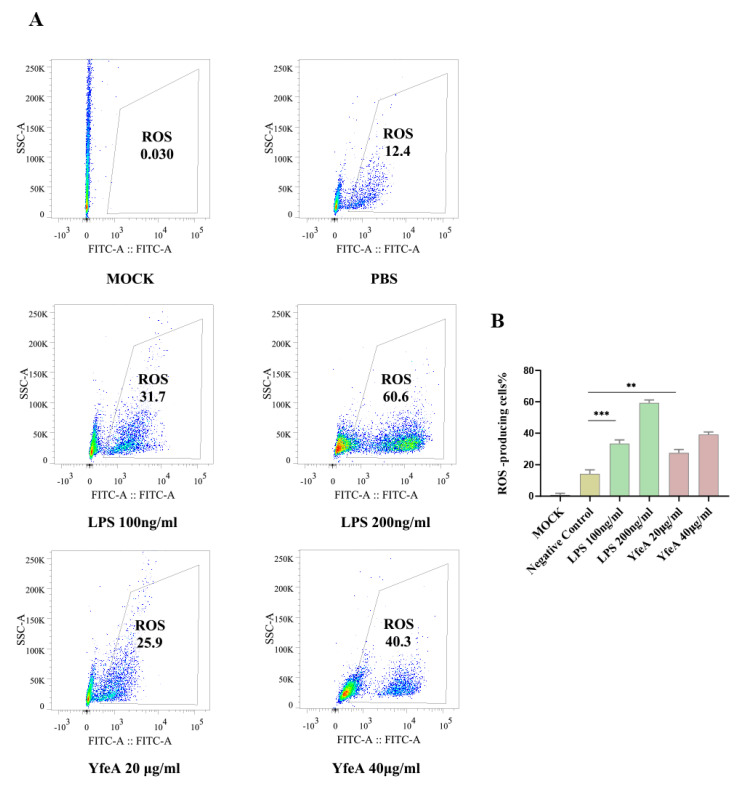
rYfeA protein triggered intracellular level of ROS. (**A**) The FCM detection of the levels of ROS induced by YfeA (20 and 40 μg/mL) and LPS (100 and 200 ng/mL). (excitation wavelength: 488 nm, emission wavelength: 525 nm, a FITC-range monitoring scope). RAW 264.7 were plated into 6-well plates at a density of 2 × 10^6^ cells/well overnight and then stimulated with LPS or YfeA for 12 h prior to ROS measurements. The DCFH-DA probe detected the intracellular ROS level. After being oxidized by ROS, DCFH-DA transformed to DCF, with a green fluorescence upon irradiation of a 488 nm laser. The fluorescence intensity of DCF was used to evaluate intracellular oxidative stress. (**B**) Statistical results of ROS levels in each treatment-group. Significant differences between groups are indicated by * *p* < 0.01 and ** *p* < 0.001.

**Figure 6 ijms-23-09627-f006:**
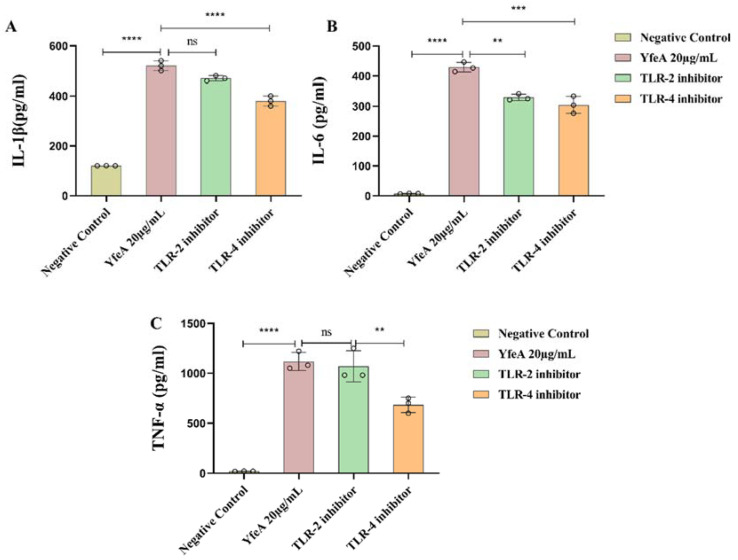
Analysis of the recognition receptor of YfeA. RAW 264.7 cells were plated into 6-well plates at a density of 2 × 10^6^ cells/well overnight, then anti-TLR2 and anti-TLR4 monoclonal antibodies were used to block their corresponding toll-like receptor. Both antibodies were added to a final concentration of 10 µg/mL for 2 h separately before YfeA stimulation. Secretion levels of (**A**) IL-1β, (**B**) IL-6, and (**C**) TNF-α in the culture supernatants were detected after 12 h incubation. Significant differences between groups are indicated by ** *p* < 0.01, *** *p* < 0.001, and **** *p* < 0.0001, ns *p* < 0.05.

**Figure 7 ijms-23-09627-f007:**
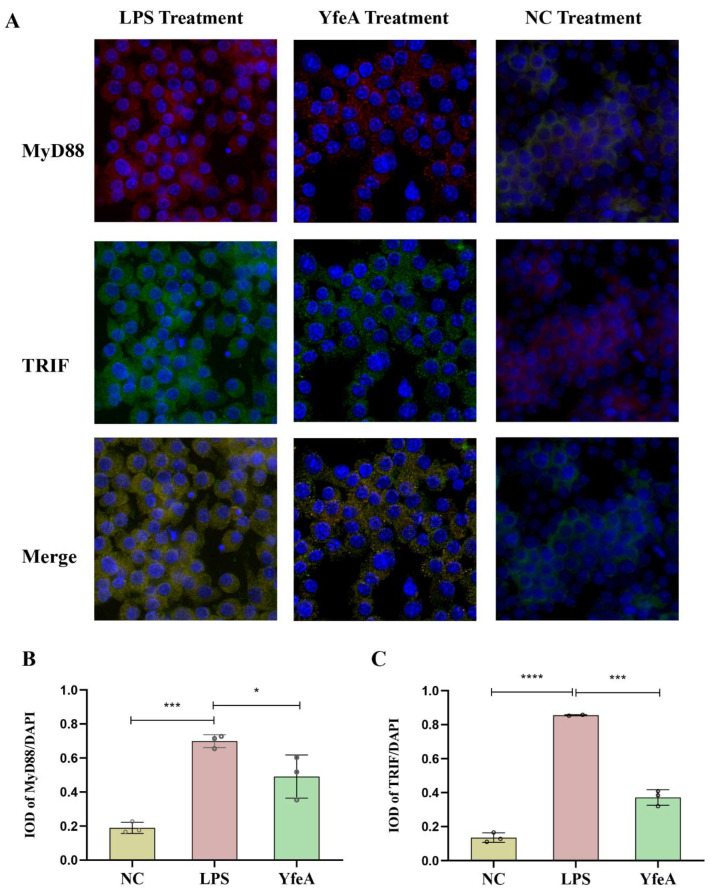
Detection of the adaptor protein TRIF and MyD88 in RAW 264.7 cells. RAW 264.7 macrophages were incubated with 20 µg/mL YfeA protein and 100 ng/mL LPS (positive control) overnight, as well as single culture media (negative control). (**A**) Indirect immunofluorescence assay. DAPI: blue light. TRIF: green light; MyD88: red light merged with yellow light. (**B**) IF analysis (40×) and statistical analysis of IOD of MyD88/DAPI. (**C**) IF analysis (40×) and statistical analysis of IOD of TRIF/DAPI. Significant differences between groups are indicated by * *p* < 0.05, *** *p* < 0.001, and **** *p* < 0.0001.

**Figure 8 ijms-23-09627-f008:**
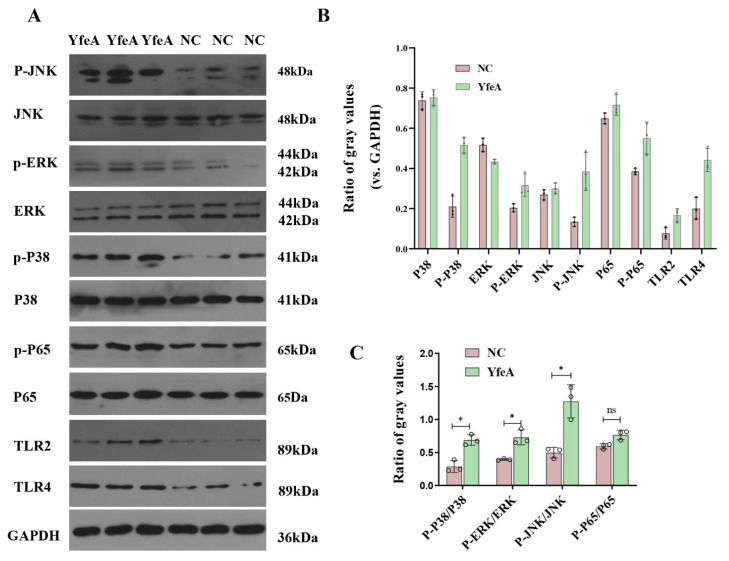
Detection of the signaling molecule involved in pro-inflammatory cytokine secretion. RAW 264.7 were plated into 6-well plates at a density of 2 × 10^6^ cells/well overnight, then pre-incubated with 20 µg/mL YfeA protein or DMEM for 12 h. The total protein was used to detect signaling molecules in MAPK and NK-κB signaling pathways. (**A**) Western blot analysis. (**B**) Gray intensity. (**C**) Ratios between phosphorylated activated signal molecules and signal molecules. Significant differences between groups are indicated by * *p* < 0.05, ns *p* < 0.05.

**Figure 9 ijms-23-09627-f009:**
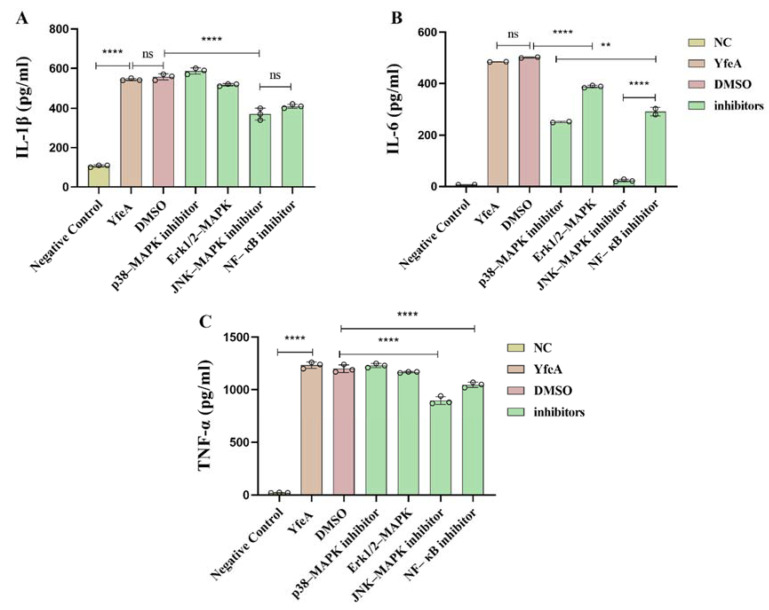
Effects of inhibitor on the pro-inflammatory cytokine secretion induced by YfeA. RAW 264.7 were plated into 6-well plates at a density of 2 × 10^6^ cells/well overnight, then SB203580 (p38-MAPK inhibitor), U0126 (Erk1/2 inhibitor), SP600125 (JNK inhibitor), and PDTC (NF-κB inhibitor) were used to block their corresponding MAPK and NF-κB signal pathways. Secretion levels of IL-1β (**A**), IL-6 (**B**), and TNF-α (**C**) in the culture supernatants were determined by ELISA. Significant differences between groups are indicated by ** *p* < 0.01 and **** *p* < 0.0001, ns *p* < 0.05.

**Figure 10 ijms-23-09627-f010:**
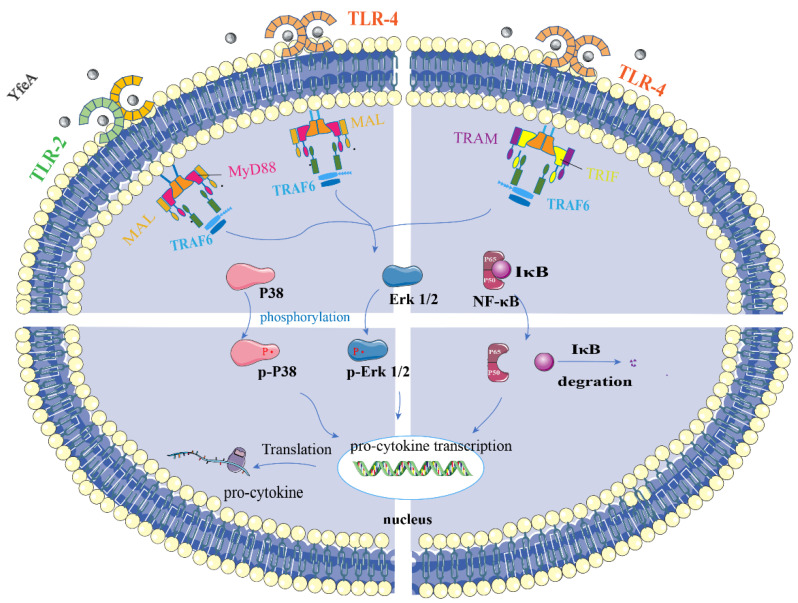
Schematic representation of the signal pathway induced by YfeA based on the above results. YfeA of *G. parasuis* triggers activation of TLR2 and TLR4, and subsequent recruitment of adaptors MyD88 and TIRAP, which in turn recruits and activates TRAF6 and other downstream signal molecules. The process leads to IκB proteasomal degradation followed by nuclear translocation of p50 and p65 and phosphorylation of p38 and ERK1/2. The activation of NF-κB and MAP kinase signaling pathways induce transcription and translation of various pro-inflammatory cytokines, including IL-1β, IL-6, and TNF-α.

**Table 1 ijms-23-09627-t001:** Primers were used in this study.

PCR Primers	Primer Sequences (50→30)	Products (bp)
**YfeA Expression Primers**		
P1 (YfeA-pET28a-F)	cagcaaatgggtcgcggatccCAGCAGTTTAAAGTGGTCACCAC	
P2 (YfeA-pET28aR)	gtggtggtggtggtgctcgagTTATTTTTCAAATCCAGCAACAAT	819
**qPCR primers**		
P3 (GAPDH-F)	GTGTTCCTACCCCCAATGTG	
P4 (GAPDH-R)	CATCGAAGGTGGAAGAGTGG	189
P5 (IL-1β-F)	GGGCCTCAAAGGAAAGAATC	
P6 (IL-1β-R)	TACCAGTTGGGGAACTCTGC	183
P7 (IL-4-F)	TCTTGATAAACTTAATTGTCTCT	
P8 (IL-4-R)	GCAGGATGACAACTAGCTGGG	160
P9 (IL-6-F)	GGGACTGATGCTGGTGACAA	
P10 (IL-6-R)	TCCACGATTTCCCAGAGAACA	147
P11 (TNF-α-F)	CGTCAGCCGATTTGCTATCT	
P12 (TNF-α-F)	CTTGGGCAGATTGACCTCAG	184

## Data Availability

The accession numbers of the *yfeA* gene of *G. parasuis* and *Yersinia pestis* are CP001321.1 and U50597.1, respectively. The PDB ID of the YfeA protein of *Yersinia pestis* is 6Q1D.

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
