# Peer review of "Metal Ion Periplasmic-Binding Protein YfeA of Glaesserella parasuis Induces the Secretion of Pro-Inflammatory Cytokines of Macrophages via MAPK and NF-κB Signaling through TLR2 and TLR4"

_ijms, 2022, doi:10.3390/ijms23179627_

Round 1

Reviewer 1 Report

Gram-negative bacteria Glaesserella parasuis (formerly Haemophilus parasuis) are the causative agents of a severe infectious disease of pigs, Glaesser's disease, which in most cases leads to the death of animals and thus causes great damage to the economy of various countries. This work is aimed at elucidating the mechanism of induction of innate immunity by the G. parasuis periplasmic protein YfeA, which is considered as a potential component of a vaccine against this dangerous infection. Therefore, this work is of undoubted interest from both practical and theoretical points of view. However, MS ijms-1843922 has a few grammatical errors that need to be corrected:

 Title. Change the initial capital letter in the word macrophage to lowercase. The Latin word “via”  is written in italics.

Lines 19 and 39. Delete “(G.parasuis)” after Glaesserella parasuis. The generic name is abbreviated to its first letter with a dot after the first mention of the full name of the species. Further, only the abbreviated name of the species is used.

Line 21. Introduce space between G. and parasuis in “G.parasuis”. Check the spaces though all the text.

Line 24. Introduce abbreviation TLR after “toll-like receptors”.

 Line 62. Correct “Salmonella eenterica” for “Salmonella enterica”

Line 69. Introduce abbreviation LPS after the word “lipopolysaccharide”.

Line 90 “proinflammatory macrophages” -? Delete “proinflammatory”.

Lines 94-95. Correct the sentence “In our study, we concentrate on the immunoprotection of YfeA in G. parasuis. macrophages “ . The meaning is not clear. The end of the sentence is incomprehensible, besides, there is no dot at the end.

Line 109.etc.” is written in italics with a dot.

Line 114. Introduce the abbreviation “SBP”.

Line 116. “G.parasuis and G.parasuis and Y.pestis” – There are not enough spaces in the names of bacteria. The word “of” after “YfeA is superfluous.

Lines 115, 120, 218, 263 – The first letter of “Figure” must be capital.

Lines 138, 355: “in vitro”, “in vivo” should be written in italics.

Line 141. Abbreviation WB should be introduced initially (Western blotting (WB)), and then it must be used (line 142 etc.).

Lines 141-144. Specify figure numbers: SDS-PAGE (Figure 2A), Western blotting (Figure 2B).

Line 145. “…endotoxin (LPS?)…”

Line 156. The sentence must begin with a capital letter.

Figure 2. In the legend, it is necessary to note that А and В show electrophoregrams obtained by SDS-PAGE and Western blotting, respectively.

Line 366. Substitute “protective” for “protectivity”.

Line 382. Substitute “Fig 10” for “Figure 10”.

An overall notation to the text. The abbreviation should be introduced once after the first mention of the word or phrase. Then, it is necessary to use this abbreviation through the text. Pay attention to it, please.

Author Response

Comments and Suggestions for Authors

This work is aimed at elucidating the mechanism of induction of innate immunity by the G. parasuis periplasmic protein YfeA, which is considered as a potential component of a vaccine against this dangerous infection. Therefore, this work is of undoubted interest from both practical and theoretical points of view. However, MS ijms-1843922 has a few grammatical errors that need to be corrected.

Ans: Thank you for your positive evaluation of our manuscript. In accordance with your comments, we have made extensive revisions to our manuscript.

- Title. Change the initial capital letter in the word macrophage to lowercase. The Latin word “via”  is written in italics.

Ans: Thanks for your kind reminder. We have changed the initial capital letter in the word macrophage to lowercase and turned the word via into italics. Please see the line: 4.

- Lines 19 and 39. Delete “(G.parasuis)” after Glaesserella parasuis. The generic name is abbreviated to its first letter with a dot after the first mention of the full name of the species. Further, only the abbreviated name of the species is used.

Ans: Thanks for your suggestion. We have deleted “(G.parasuis)” after “Glaesserella parasuis”. Please see Lines 19 and 39.

- Line 21. Introduce space between G. and parasuis in “G.parasuis”. Check the spaces though all the text.

Ans: We have introduced space between G. and parasuis in “G.parasuis” in the text. Please see lines 21, 116, 122, 123, 350, 375, and 376.

- Line 24. Introduce abbreviation TLR after “toll-like receptors”.

Ans: We have introduced the abbreviation TLR after “toll-like receptors.” Please see the line: 24. Thanks.

- Line 62. Correct “Salmonella eenterica” for “Salmonella enterica”

Ans: Thanks for your kind reminder. We have corrected “Salmonella eenterica” for “Salmonella enterica”. Please see the line: 61.

- Line 69. Introduce abbreviation LPS after the word “lipopolysaccharide”.

Ans:. We have written the full name of LPS in our manuscript as per your suggestion. Please see line: 69.

- Line 90 “proinflammatory macrophages” -? Delete “proinflammatory”.

Ans: We have deleted the word “proinflammatory”. Please see line: 90. Thanks.

- Lines 94-95. Correct the sentence “In our study, we concentrate on the immunoprotection of YfeA in G. parasuis. macrophages “ . The meaning is not clear. The end of the sentence is incomprehensible, besides, there is no dot at the end.

Ans: We deleted the word “macrophage” at the end of the sentence, and then the meaning of this sentence became clear and understandable. Please see lines: 94-95.

- Line 109. “etc.” is written in italics with a dot.

Ans: We have written the “etc.” in italics with a dot. Please see line: 109.

- Line 116. “G.parasuis and G.parasuis and Y.pestis” – There are not enough spaces in the names of bacteria. The word “of” after “YfeA” is superfluous.

Ans: Thanks for your kind reminder. We have introduced space in “G.parasuis and Y.pestis”, and deleted the word “of” after “YfeA”. Please see line: 116.

- Lines 115, 120, 218, 263 – The first letter of “Figure” must be capital.

Ans: We have Capitalized the first letter of “figure”. Please see lines: 115, 121, 222 and 268.

- Lines 138, 355: “in vitro”, “in vivo” should be written in italics.

Ans: We have written “in vitro”, “in vivo” in italics. Please see lines:139 and 362.

- Line 141. Abbreviation WB should be introduced initially (Western blotting (WB)), and then it must be used (line 142 etc.).

Ans: Thanks for your kind reminder. We have written the full name of WB in our manuscript as per your suggestion. Please see line: 144.

- Lines 141-144. Specify figure numbers: SDS-PAGE (Figure 2A), Western blotting (Figure 2B).

Ans: We have specified figure numbers: SDS-PAGE (Figure 2A), and Western blotting (Figure 2B). in our manuscript as per your suggestion. Please see line: 144-145. Thanks.

- Line 145. “…endotoxin (LPS) 

Ans: We have introduced LPS, the main component of endotoxin, after “endotoxin”. Please see line: 146.

- Line 156. The sentence must begin with a capital letter.

Ans: We have capitalized the first letter of “we”. Please see line: 157.

- Figure 2. In the legend, it is necessary to note that А and В show electrophoregrams obtained by SDS-PAGE and Western blotting, respectively.

Ans: Thanks for your kind reminder. We have specified that Figure 2A was obtained by SDS-PAGE. Please see lines: 150-151.

- Line 366. Substitute “protective” for “protectivity”.

Ans: We have substituted “protective” for “protectivity” as per your suggestion. Please see line: 373. Thanks.

- Line 382. Substitute “Fig 10” for “Figure 10”.

Ans: We have substituted “Fig 10” for “Figure 10”. Please see line: 389.

Overall, we appreciate your warm work earnestly. Thanks.

Reviewer 2 Report

The manuscript of Yang et al., entitled “Metal ion periplasmic-binding protein YfeA of Glaesserella 2 parasuis induces the secretion of pro-inflammatory cytokines of Macrophage via MAPK and NF-κB signaling through TLR-2 and TLR-4” showed that YfeA could be a potent candidate for vaccine development against  G. Parasuis. They have used in vivo mouse model to identify the protection provided by the vaccine candidate. While in vitro study in Raw 264.7 cells was used to find out the signaling pathways involved. This is a very well-articulated article. However, some concerns needed to be addressed before publication. Please find below the minor concerns of this study.

Comments:

1.     Authors have represented the survival of mice in fractions however in the discussion they have explained the result in percentage survival. Thus they are suggested to represent figure 3 in percentage survival instead of fractional survival.

2.     In Fig 3A, rYfeA alone/PBS/Adjuvant alone showed 100% mortality at 48h however, it is written 36h (line 159). Thus this should be corrected.

3.     In line 162, instead of stating “nearly half of the mice”, they should provide the exact percentage of mice mortality.

4.     The author mentioned that after passive immunization, all the survived mice got recovered. Thus it would be interesting if the author could plot a graph of weight loss of mice and show the weight gain in recovered mice.

5.     Line 218, replace fig 2D with Fig 4D.

6.     In Figure 8, the band of pERK looks less resolved than the total ERK (the gap between 44 n 42kDa protein is more). Thus the author should explain whether they have used the same WB membrane to probe pERK and total ERK or a different membrane. If they have used a different membrane, then they should probe the same membrane for both phosphoprotein and total protein and quantify again.

7.     The size of pP65 and p65 should be approximately 65kDa however, figure 8A shows 60 and 64 kDa, respectively. Thus the authors are suggested to correct this and also provide all antibodies list with detailed information.

8.     Quantifying and representing protein bands with respect to housekeeping protein (GAPDH) would be more informative and interesting than just plotting gray values. Thus they are suggested to replot Figure 8B.

9.     Line 375, the authors are suggested to provide additional information in the supplemental section instead of writing data not shown.

Author Response

Comments and Suggestions for Authors

Yang and colleagues have used in vivo mouse model to identify the protection provided by the vaccine candidate. While in vitro study in Raw 264.7 cells was used to find out the signal pathways involved. This is a very well-articulated article. However, some concerns needed to be addressed before publication. Please find below the minor concerns of this study.

Ans: We are grateful for your positive evaluation. We have made extensive revisions to our manuscript and supplemented extra data to make our results clear, per your editorial comments.

Comments:

  1. Authors have represented the survival of mice in fractions however in the discussion they have explained the result in percentage survival. Thus they are suggested to represent figure 3 in percentage survival instead of fractional survival.

Ans: Thanks for your pointing out this problem. We have represented figure 3 in percentage survival instead of fractional survival. Please see a new figure 3A in line: 181.

  1. In Fig 3A, rYfeA alone/PBS/Adjuvant alone showed 100% mortality at 48h however, it is written 36h (line 159). Thus this should be corrected.

Ans: Thanks for your kind reminder. rYfeA alone/PBS/Adjuvant alone showed 100% mortality at 48h in Fig 3A. We have corrected it. Please see line: 161.

  1.  In line 162, instead of stating “nearly half of the mice”, they should provide the exact percentage of mice mortality.

Ans: We have written the exact percentage of mice:40%, instead of stating “nearly half of the mice”. Please see line: 163. Thanks.

  1. The author mentioned that after passive immunization, all the survived mice got recovered. Thus it would be interesting if the author could plot a graph of weight loss of mice and show the weight gain in recovered mice.

Ans: Thank you for your helpful comments. Unfortunately, because the mice challenge experiments lasted only 72 hours, we didn’t record the mice's body weight change in the experiment. This is a great suggestion.

  1.  Line 218, replace fig 2D with Fig 4D.

Ans: Thanks for your kind reminder. We have replaced fig 2D with Fig 4D. Please see line: 222. Thanks.

  1. In Figure 8, the band of pERK looks less resolved than the total ERK (the gap between 44 n 42kDa protein is more). Thus the author should explain whether they have used the same WB membrane to probe pERK and total ERK or a different membrane. If they have used a different membrane, then they should probe the same membrane for both phosphoprotein and total protein and quantify again.

Ans: We used the same WB membrane to probe pERK and total ERK. Protein phosphorylation levels are more challenging to detect than total protein levels using WB assay, especially when probed at the same secondary antibody dilution ratio and developed with the same exposure time. Not only pERK, but also p38 and p65 are more challenging to detect than their corresponding total protein.

7.The size of pP65 and p65 should be approximately 65kDa however, figure 8A shows 60 and 64 kDa, respectively. Thus the authors are suggested to correct this and also provide all antibodies list with detailed information.

Ans: Thanks for your kind reminder. We checked the observed band sizes of the pP65 and p65, and they both are 65 kDa as you pointed out. Therefore, we have corrected the band size of pP65 and p65. Please see Fig 8A. In addition, we have provided all antibody lists with detailed information. Please see lines: 502-509.

  1. Quantifying and representing protein bands with respect to housekeeping protein (GAPDH) would be more informative and interesting than just plotting gray values. Thus they are suggested to replot Figure 8B.

Ans: Thanks for your kind reminder. Fig 8B showed the ratio of the gray value of target protein bands vs the gray value of GAPDH. We have corrected the legend of the Y-axis of Fig 8B. Please see a new figure 8B in line: 320.

  1. Line 375, the authors are suggested to provide additional information in the supplemental section instead of writing data not shown.

Ans: Thanks for your suggestion. As illustrated in Fig 1A, we used SDS-PAGE and WB to detect the purified pET-32a-His protein (MW:19 Kda). As shown in Fig 2B-D, we utilized the pET-32a-His protein as a negative control, and the rYfeA-treated group showed a higher transcription level of pro-inflammatory cytokines.

Figures are in attachment. Please see the attachment.

Overall, we appreciate your warm work earnestly. Thanks.

Round 2

Reviewer 2 Report

Thank you so much for making corrections.

All the best.